# Multi-Objective Function Optimization of Cemented Neutralization Slag Backfill Strength Based on RSM-BBD

**DOI:** 10.3390/ma15041585

**Published:** 2022-02-20

**Authors:** Mingqing Huang, Lin Chen, Ming Zhang, Shulin Zhan

**Affiliations:** 1Zijin School of Geology and Mining, Fuzhou University, Fuzhou 350108, China; seango@fzu.edu.cn (M.H.); n191620002@fzu.edu.cn (M.Z.); forest_zh@163.com (S.Z.); 2State Key Laboratory of Comprehensive Utilization of Low-Grade Refractory Gold Ores, Longyan 356214, China

**Keywords:** neutralization slag with high mud content, Box–Behnken design, response surface method (RSM), scanning electron microscope (SEM), multi-objective optimization

## Abstract

Tailings produced in the beneficiation of Carlin-type gold deposits are characterized by fine particle size and high mud content. When neutralized with wasted acid generated by pressurized pre-oxidation, the tailings turn to neutralized slag and perform as a novel backfill material. To understand the influential behavior of variable factors on the strength and its optimization of cemented neutralization slag backfill, RMS-BBD design test was carried out with 56–60% slurry mass fraction, 12.5–25% cement/(neutralization slag + waste rock) (i.e., C/(S+R)) and 30–40% waste rock content. A modified three-dimensional quadratic regression model was proposed to predict the strength of cemented neutralization slag backfill. The results showed that backfill strength predicted by the modified ternary quadratic regression model was in high coincidence with the data of backfill mixture tests. C/(S+R) was predominant in backfill strength with regard to every single influential factor throughout the curing age, and the mass fraction of slurry had a significant effect on the later strength. From the perspective of economic and engineering operation, a multi-objective function method was further introduced to optimize the backfill strength. The optimal mixture proportion of cemented neutralized slag backfill slurry was: 58.4% slurry mass fraction, 32.2% waste rock content, and 20.1% C/(S+R). The backfill strength of this mixture proportion on days 7, 28 and 56 was verified as 0.42, 0.64 and 0.85 MPa, respectively. RSM-BBD design and multi-objective function optimization proposed a reliable way to evaluate and optimize the strength of neutralized slag backfill with high mud content.

## 1. Introduction

Backfill mining has been highly valued by the mining industry as an important approach to fulfill green mining. At present, cemented backfill mining technology not only effectively prevents the potential surface subsidence by filling underground mined-out areas but also alleviates the environmental pollution caused by abundant solid wastes such as waste rocks and tailings [1,2,3,4,5]. Since the cement material always accounts for a relatively high proportion of backfill operating costs, it is a key issue to maintain an equilibrium between backfill cost and backfill strength [6]. To control the backfill operating cost based on the required backfill strength, scholars worldwide have committed to selecting materials and optimizing the mixture proportion of backfill for the past few decades, conducting in-depth research on its mechanical response and microstructure evolution. Deng et al. [7] took tailings, ordinary Portland cement, and different additives as backfill materials, and found the strength evolution law of cement tailings backfilling under different curing years. Wang et al. [8] tested the strength of new slurry filling materials through coal zircon, laterite, and cement, obtained the best ratio between materials, and reduced the filling cost. Rakhimova et al. [9] considered the micro results of backfill and found that the close combination of limestone and binder produced a dense physical structure, which improved the compressive strength of sodium carbonate slag cement slurry. Kopwade-Patil et al. [10] investigated the early and late curing effects of hardened cement paste formed by pozzolanic cement and ordinary Portland cement.

When new materials are introduced in the backfill, it is necessary to consider whether the backfill strength can meet the requirements. The strength and economic benefit of backfill are important factors to be considered in the mine production and application, which are closely related to the optimal proportioning scheme. The response surface method (RSM) [11] is a widely used experimental design method to optimize the test scheme, reduce the planned experiment number, and fit the relationship between factors within the range and response results [12,13,14]. At present, the RSM method has been widely used in process optimization design [15,16,17,18,19,20]. The strength of the backfill body is an important parameter in the backfill design, especially when the backfill body plays a supporting role to ensure the subsequent safe mining [21,22,23,24]. Some scholars optimized the mixture proportion of backfill materials by using RMS to study the influential behaviors of variable factors and their interaction on backfill strength at different curing ages [25,26,27]. Among massive backfill parameters, backfill strength is critical to maintaining the stability of stopes and rock mass adjacent to the mined-out areas. Under the condition of constraint strength, the proportion of backfill materials can be optimized by using the multi-objective function method to obtain the best economic benefit [28].

The Carlin-type gold mine contributes to about 10% of gold production worldwide in recent years. This kind of deposit is characteristic in micro disseminated minerals, which have to be ground to extra fine particles, and subjected to pressurized oxidation to separate the target minerals from host rocks during mineral processing. However, this specific technique results in a large amount of acid wastewater. A traditional and mature treatment of acid wastewater is two-step neutralization by limestone and lime powder. Unfortunately, the cost of traditional two-step neutralization is as high as CNY 90 per ton of gold concentrate. Shuiyindong gold mine, a typical Carlin-type gold deposit in southwest China [29], finds that alkaline tailings are a successful substitute for limestone to neutralize the acid wastewater. After neutralization, the slurry is pumped to the thickener for solid–liquid separation, and the thickener underflow is called neutralization slag. Though the neutralization slag has been used as backfill material in the Shuiyindong gold mine, its slurry concentration and backfill strength are hardly satisfied due to its high mud proportion. The proper addition of waste rock as coarse aggregate may be a promising way to overcome the above problems, yet the size, proportion, and effectiveness of waste rock in backfill mixture proportion stay indistinct.

The objective of this work is to explore the slurry concentration and backfill strength evolution roles of cemented neutralization slag backfill in the presence of waste rocks. Firstly, waste rocks were added to neutralization slag slurry to test the improvement of backfill slurry concentration. Secondly, the response surface regression model (RSM) for different backfill curing ages was established by using Box–Behnken design (BBD). Besides, the effects of slurry mass fraction, waste rock proportion (neutralization slag/waste rock), cement/(neutralization slag + waste rock) (C/(S+R)), and their interactions on backfill strength at different curing ages were concluded. Thirdly, the backfill mixture proportion of cemented neutralization slag was optimized by multi-objective programming towards high slurry mass fraction and backfill strength. Thus, a backfill strength optimization method will be proposed with regard to cemented neutralization slag backfill based on RSM-BBD and multi-objective function optimization.

## 2. Materials and Methods

### 2.1. Materials

The materials included neutralization slag with high mud content (fine aggregate), waste rock (coarse aggregate), and M32.5 Portland cement (cementing agent). All materials were supplied by Shuiyindong gold mine in Zhenfeng County, Guizhou Province, China. The particle size distribution range of neutralization slag was 1.035–683.912 μm, where *d*_10_ = 2.47 μm, *d*_30_ = 4.52 μm, *d*_60_ = 9.23 μm, *C*_u_ was 3.74, and *C*_C_ was 0.896. The physical parameters and chemical compositions of neutralization slag measured by X-ray fluorescence spectrometry (XRF) are shown in Table 1 and Table 2, respectively. The particle size distribution of neutralization slag measured by focused beam reflection measurement (FBRM) is shown in Figure 1. The full-size waste rocks from mine waste dumps were crushed to −5 mm (Table 1). The particle size distribution of crushed waste rock measured by the screening method is shown in Figure 2.

### 2.2. Determination of Slurry Parameter Range

The acidity of the neutralization slag slurry was measured by a starter 2100 pH meter (Ohaus instrument, Pine brook, New York, NY, USA) equipped with an st310 electrode. The measurement showed that the neutralization slag slurry has a pH range of 8.6–8.8, which is suitable for backfill aggregate. The neutralization slag with high mud content was used as a single aggregate, and the mass fraction of the slurry was only about 45%. When the mass fraction of slurry was higher than 45%, the fluidity of slurry becomes poor and the viscosity was very strong. In this test, a certain proportion of waste rock was added to neutralization slag to improve slurry concentrations. With a standard specification slump bucket that 100 mm in the upper opening, 200 mm in the lower opening, and 300 mm in height, the slurry expansion with a mass fraction of 64%, waste rock content of 25%, and C/(S+R) of 33% was measured to be 27.2 cm. The fluidity was too poor (Figure 3) to transport to underground mined-out areas by gravity. Hence, from the perspective of slurry fluidity, the slurry mass fraction was limited to 60%, and the range values of various parameters were determined as follows: slurry mass fraction 56–60%, waste rock content 30–40%, and C/(S+R) 12.5–25%.

### 2.3. Preparation of Cemented Backfill

The Box–Behnken design scheme was introduced to optimize the mixture proportion of cemented neutralization slag backfill. The influence of various factors on the backfill strength was mainly investigated, and the relationship between evaluation response models was preliminarily established. The test took slurry mass fraction (*X*_1_), waste rock content (*X*_2_), and C/(S+R) (*X*_3_) as input variables. Three factors and three levels were designed according to the design principle (Table 3). Each factor has a low-level value (−1), intermediate value (0), and advanced value (+1). Table 4 shows the test design matrix of coding and non-coding units, and the software runs 17 batches.

All samples were prepared according to the following steps (Figure 4): dry neutralization slag and waste rock were weighed and mixed in the mixer until the slurry was evenly distributed. Prepared slurry was uniformly loaded into the 70.7 m× 70.7 m× 70.7 m molds and slightly vibrated for 10 s to prevent cracks and bubbles within the backfill. All samples were demolded within 36 h and then cultured in a standard curing room (YH-40B) (temperature: 25 ± 2 °C, humidity ≥ 95%) for 7, 28, and 56 days, respectively. The compressive strength of cemented backfill was tested according to the national standard test method for mechanical properties of ordinary concrete (GB/T 50081-2002). The main mechanical machine was QKX-ZSZ-4000 Rock mass true triaxial dynamic and static load test system (Qingdao qiankunxing intelligent Co., Ltd., Qingdao, China).

### 2.4. Statistical Analysis

RSM is used to optimize the mix proportions with satisfactory backfill strength. RSM is a method based on ternary quadratic regression function model, and the regression equation is as follows [30,31]:(1)$${Y=\alpha }_{0}+\sum_{i=1}^{k}\alpha_{i}X_{i}+\sum_{i=1}^{k}\alpha_{\mathrm{ii}}X_{i}^{2}+\sum_{i<j}\alpha_{\mathrm{ij}}X_{i}X_{j}+\varepsilon$$
where *Y* is the uniaxial compressive strength of cemented neutralization slag backfill; *X*_i_ and *X*_j_ refer to slurry parameters including the mass fraction of the slurry, waste rock content, and C/S+R; *X*_i_^2^ and *X*_i_*X*_j_ refer to secondary and interactive independent variables; α_0_ is the model intercept coefficient; k is the number of independent variables (k = 3 for the present study); α_i_, α_ii_, and α_ij_ is the coefficient of the linear term, quadratic term, and second-order term respectively; *ε* is random error. Based on the proposed model, analysis of variance was carried out to determine the influence of single or multiple factors on backfill strength. The fitting quality of the regression model was expressed by the determination coefficient (*R*^2^, adj *R*^2^), relationships between tested value and predicted value, and the residual analysis.

## 3. Results and Discussion

### 3.1. Evaluation of Strength Test and Model Fitting

All analytical tests are carried out in triplicate according to the Box–Behnken design. As shown in Table 5, *Y*_1_, *Y*_2_ and *Y*_3_ are tested values of backfill strength at 7, 28 and 56 days, respectively; *Y*_1_*, *Y*_2_* and *Y*_3_* are predicted values of backfill strength according to correlated response regression curves. The multiple regression fitting functions are shown in Equations (2)–(4).
(2)$$\begin{array}{c} Y_{1}=-29.32637+0.858937X_{1}+0.204500X_{2}+0.046460X_{3}\;-\;0.002600X_{1}X_{2}\;-\;0.000620X_{1}X_{3}\\ -\;0.000784X_{2}X_{3}\;-\;0.006344{X_{1}}^{2}\;-\;0.000545{X_{2}}^{2}+0.001053{X_{3}}^{2}\;,\;(R^{2}=0.9931) \end{array}$$
(3)$$\begin{array}{cc} Y_{2}= & -18.18050+0.665938X_{1}+0.106150X_{2}\;-\;0.454540X_{3}-0.002075X_{1}X_{2}+0.005520X_{1}X_{3}\\ & +0.000552X_{2}X_{3}\;-\;0.005563{X_{1}}^{2}\;-\;(3.53492\times 10^{-17}){X_{2}}^{2}+0.004192{X_{3}}^{2}\;,\;(R^{2}=0.9801) \end{array}$$
(4)$$\begin{array}{c} Y_{3}=-35.55125+1.07144X_{1}+0.252775X_{2}\;-\;0.161840X_{3}\;-\;0.003400X_{1}X_{2}+0.002780X_{1}X_{3}\\ -\;0.001112X_{2}X_{3}\;-\;0.008125{X_{1}}^{2}\;-\;0.000440{X_{2}}^{2}+0.002259{X_{3}}^{2}\;,\;(R^{2}=0.9813) \end{array}$$

Statistical analysis is performed using software Design-Expert 13. The significance of the regression equation is an indicator to verify the consistency of the proposed model and tested values. The significance and suitability of the selected regression model for each factor are investigated using variance analysis [32]. Correlation coefficient *R*^2^ represents the difference between the response function and the true value. The range of R^2^ is 0 to 1, and the error between the predicted value and the true value becomes low when R^2^ is approaching 1. The variance of the functional model (Table 6) shows that the *F*-value in each model is higher than *F*_0.05_(3,9) = 3.86, and the *p*-value in the three models is lower than 0.0001. The response model is significant when *p* < 0.05 [33]. The comparison of tested backfill strength and the predicted value at the age of 7, 28, and 56 days (Figure 5) show that tested results are very close to the predicted values of the polynomial response model [34]. The results also show that the parameters of the model are significant within a 95% confidence interval.

### 3.2. Influence of Single Factor on Backfill Strength

The analysis of the response function model (Table 6) shows the significant *p*-values of slurry mass fraction and C/(S+R) on backfill strength are <0.005, the correlation *F*-values are >3.86, and the *F*-value of waste rock content is low. These findings indicate that waste rock rarely participates in cement hydration reaction and has little impact on backfill strength. Comparing the correlation *F*-values of various factors on backfill strength shows that the influence degree of each factor on backfill strength is C/(S+R) (*X*_3_) > slurry mass fraction (*X*_1_) > waste rock content (*X*_2_). Figure 6 also shows that C/(S+R) (*X*_3_) dominates in backfill strength.

Backfill strength at the same curing age is positively correlated with slurry mass fraction *X*_1_ and cement–sand ratio *X*_3_, indicating that backfill strength increases with the increase of *X*_1_ or *X*_3_.

#### 3.2.1. Influence of Single Factor on Backfill Strength

The maximum *F*-value (Table 6) at 56 days indicates that the mass fraction of slurry has a significant impact on the later backfill strength. Backfill strength with lower mass fraction grows in the later stage due to continuous hydration reaction between internal water and cement. It is necessary to continuously replenish water to backfill during curing. For the slurry with a higher mass fraction, it supports the dense skeleton more effectively as a proportion of neutralizing slag content increases. Results show that the mass fraction of slurry has a great influence on later backfill strength, which is consistent with the conclusion of tested data and variance analysis.

#### 3.2.2. Effect of C/(S+R) on Backfill Strength

Neutralization slag is a byproduct of alkaline tailings and waste acid at pressured peroxidation plant. The high content of carbonate in tailings is amenable to the fine particles of neutralization slag. The main composition of neutralization slag changes from carbonate to sulfate in pressured and thermal environments, which promotes the slurry setting and cement hydration. In this case, cement not only bonds the backfill material but also affects the strength and cost of the backfill. The higher C/(S+R), the greater backfill strength at each curing period with increasing slurry mass fraction. Neutralization slag contains SO_3_ and Al_2_O_3_, and the generated sulfate may precipitate in the pores, which has a beneficial impact on the development of backfill strength [35]. Since CaSO_4_·2H_2_O grains generated by neutralization are fine, it accelerates the reaction rate of CaSO_4_·2H_2_O and C_3_A (calcium aluminate). At the same time, the hydrated calcium aluminate sulfate is generated rapidly, and thus promotes the hardening rate of slurry. The diffraction intensity of hydrated granular wollastonite is greatly improved under the excitation of sulfate, which enhances the compacting and early strengthening effect.

#### 3.2.3. Effect of Waste Rock Content on Backfill Strength

As the main “skeleton” of backfill, the inert waste rock does not participate in the cement hydration reaction. The *F*-value of backfill strength is <3.86 (Table 6), indicating that waste rock content exerts little influence on backfill strength. Macropores within the backfill increase as the proportion of waste rock increases (Figure 6). For a constant slurry mass fraction, the proportion of neutralizing slag decreases with increasing coarse aggregate content due to the total amount constraint, and thus the hydration products fail to fully fill the backfill pores. Therefore, backfill strength tends to be limited if too many waste rocks are added to the slurry.

### 3.3. Response Surface Analysis

Contour lines and response surface diagrams of backfill strength at each curing age are obtained from regression Equations (2)–(4). Considering the interactive effect of variable influential factors on backfill strength, three-dimensional graphics (Figure 7, Figure 8 and Figure 9) are used to optimize the backfill strength of neutralization slag.

#### 3.3.1. Response Surface Analysis of Backfill Strength at 7 Days

The regression model of backfill strength at 7 days indicates that the interaction of slurry mass fraction *X*_1_ and waste rock content *X*_2_ has the most significant impact on backfill strength. This interaction is described in Figure 7 when the cement–sand ratio is 18.75%. When the waste rock content is low and constant, the 7-day backfill strength increases gradually with the increase of slurry mass fraction. It indicates that backfill strength at the early stage can be improved by increasing slurry mass fraction and waste rock content simultaneously. The incomplete cement hydration of slurry is amenable for this phenomenon considering that backfill strength is mainly dependent on the initial skeleton structure. The *F*-value of *X*_2_*X*_3_ is also high since backfill in high concentration accounts for more coarse aggregate, its skeleton structure is more stable.

#### 3.3.2. Response Surface Analysis of Backfill Strength at 28 Days

The *F* value of *X*_1_*X*_3_ is highest in the 28-day response model, indicating that the interaction of slurry mass fraction and the cement–sand ratio is dominant in backfill strength at 28 days. Figure 8 shows the interaction response diagram of slurry mass fraction and cement–sand ratio when the waste rock content is 35%. When the slurry mass fraction is 60% and the cement–sand ratio increases from 12.5% to 25%, the strength of backfill increases twice as much. When the cement–sand ratio is 25%, strength backfill increases linearly with the increase of slurry mass fraction. In the middle and later curing stages, the hydration of cement fully reflects, and the C-S-H gel also increases gradually. Additionally, the cement amount raises, and the hydration products tend to build an integral dense skeleton, which is beneficial to the growth of backfill strength. This response surface analysis is consistent with variance analysis.

#### 3.3.3. Response Surface Analysis of Backfill Strength at 56 Days

Figure 9 shows the interactive effect of cement–sand ratio and waste rock content on backfill strength at 56 days when the slurry mass fraction is 58%. From the perspective of *F*-values (Table 6), cross-terms of variable influential factors exert almost the same effect on backfill strength in the later curing stage. Backfill strength reaches the maximum when the C/(S+R) increases to 25% and the waste rock content remains unchanged (Figure 9). This mixture proportion promotes a sufficient hydration reaction within the backfill, and thus a compact skeleton forms without any obvious cracks.

### 3.4. SEM Analysis of Backfill at Different Curing Ages

The SEM images of backfill at the curing age of 7, 28, and 56 days are shown in Figure 10, respectively. At 7 days, only parts of cement hydration reaction occur resulting in less hydration of C-S-H gel, and aggregate and cementitious materials are not completely cohesive, and obvious cracks and raw materials are clearly attached to the sample surface (Figure 10a). At 28 days, the columnar ettringite crystalline structure (AFt) is formed in the backfill body. With a small amount of calcium silicate gel generated, the hydration products gradually fill the gap between the waste rock particles and the neutralized slag particles. Thus, the density of the inner microstructure of the backfill body increases, cracks, and porosity shrink, while the backfill becomes dense (Figure 10b). At 56 days, further diffusion of hydration products permeates the entire skeleton space and forms a complete macro support network system, the gel effect is strong enough to form a compact skeleton structure, which indicates that the neutralization slag is fully involved in the hydration reaction. It is conducive to optimizing the microstructure of the backfill and improves the integral strength of cemented neutralization slag backfill. SEM analysis of backfill at different curing ages also supports the results of response models.

### 3.5. Backfill Strength Optimization and Verification

The multi-objective function optimization method is used to optimize the slurry ratio. The ultimate objective is to select a mixture proportion that achieves equilibrium between mechanical property and operating cost of backfill. According to the slurry mass fraction (*X*_1_), waste rock content (*X*_2_) and C/(S+R) (*X*_3_), the mass of neutralizing slag *M_a_*, waste rock mass *M_s_*, cement mass *M_j_*, and water mass *M_w_* per cubic meter slurry are calculated, respectively. In stage open stoping with subsequent backfill method, the 7-day(*Y*_1_), 28-day(*Y*_2_) and 56-day(*Y*_3_) backfill strength is usually designed greater than 0.3, 0.5, and 0.8 MPa, respectively. Hence, taking backfill strength as the constraint condition, the optimization model is established as shown in Equation (5).
(5)$$f=1.2M_{a}{+17M}_{s}{+380M}_{j}+1.6M_{w},$$
(6)$$\frac{M_{a}}{2.418}+\frac{M_{s}}{2.67}+\frac{M_{j}}{3.0}{+M}_{w}=1,$$
(7)$$\frac{M_{a}{+M}_{s}{+M}_{j}}{M_{a}{+M}_{s}{+M}_{j}{+M}_{w}}{=X}_{1},$$
(8)$$\frac{M_{s}}{M_{a}{+M}_{s}}{=X}_{2},$$
(9)$$\frac{M_{j}}{M_{a}{+M}_{s}}{=X}_{3},$$
where *f* is the operation cost of cemented neutralization slag backfill, CNY/m^3^; the costs of neutralization slag, waste rock, cement, and water are 1.2, 17, 380 and 1.6 CNY/ton, respectively.

A calculation model can be derived by combining Equations (2)–(9). The C++ program language is used for model solving, and its programming results are listed in Table 7.

When the mass fraction of slurry is 58.4%, the waste rock content is 32.2% and the cement–sand ratio is 20.1% with a minimum operating cost of 63.1 CNY/m^3^. Backfill strength at the age of 7, 28 and 56 days is 0.46, 0.65 and 0.81 MPa, respectively, which all meet the designed strength requirements. To test the reliability of this optimization, nine modules of backfill with 58.4% slurry concentration, 32.2% waste rock content, and 20.1% cement–sand ratio are prepared and cultured in standard curing environments. The average uniaxial compressive strength of three backfill samples at the age of 7, 28 and 56 days is 0.42, 0.64 and 0.85 MPa, respectively (Table 8). The error between the test value and the predicted value is basically within the 95% confidence interval, which proves that the multi-objective programming method is reliable.

## 4. Conclusions

In this study, the neutralization slag of the Carlin-type gold mine was applied to the backfill of goaf for the first time. The slurry proportion test was carried out by the RSM-BBD method, and the response model was established for the strength of cemented backfill at 7, 28 and 56 days. The F- and *p*-values showed that the response model had high significance, which could simulate the development of the strength of this celestial body.The research showed that the strength of backfill at the same curing age was positively correlated with the slurry mass fraction *X*_1_ and the cement–sand ratio *X*_3_. The cement–sand ratio had a significant impact on the strength of the backfill, while the slurry mass fraction had a great impact on the later strength of the backfill.The interaction between slurry mass fraction and waste rock content had a great impact on the early strength of backfill. Additionally, the interaction between slurry mass fraction and cement–sand ratio had a significant impact on the middle and late strength of the backfill.SEM analysis results showed that with the increased curing time, the cement hydration reaction was sufficient and a large number of C-S-H cementitious molecules were generated. Thus, the internal skeleton of the backfill became dense, and the strength enhancement effect was remarkable.The optimal ratio was obtained: the mass fraction of slurry was 58.45%, the content of waste rock was 32.17%, and the cement–sand ratio was 20.13%. Through the confirmatory test, the results showed that the strength of the backfill at 7, 28 and 56 days was 0.42, 0.64 and 0.85 MPa, respectively, meeting the requirements of the target strength.

## Figures and Tables

**Figure 1 materials-15-01585-f001:**
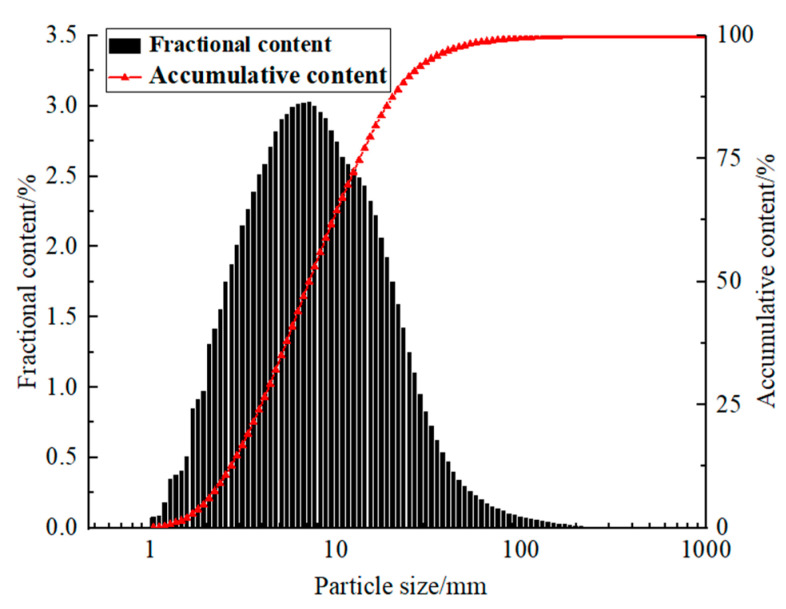
Particle size composition of neutralizing slag in backfill aggregate.

**Figure 2 materials-15-01585-f002:**
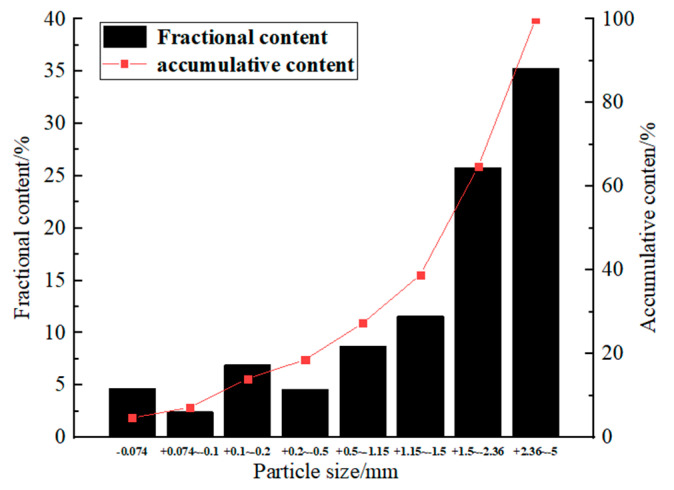
Particle size composition of waste rock filled with aggregate.

**Figure 3 materials-15-01585-f003:**
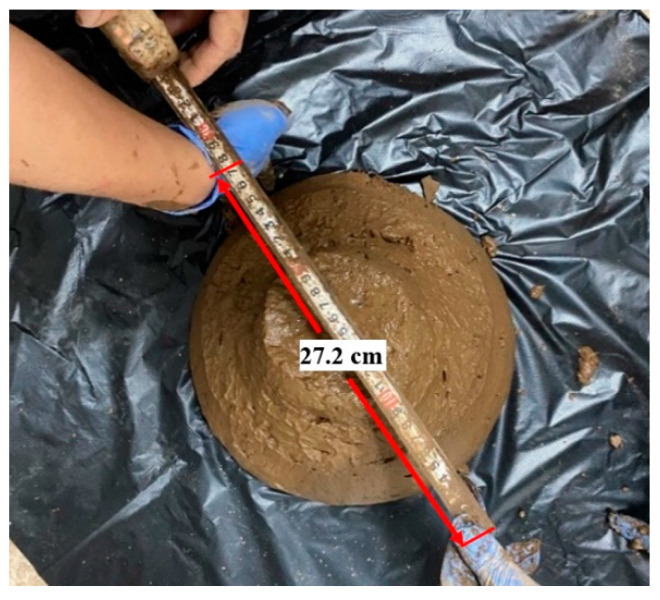
Measurement of slurry expansion with a mass fraction of 64%.

**Figure 4 materials-15-01585-f004:**
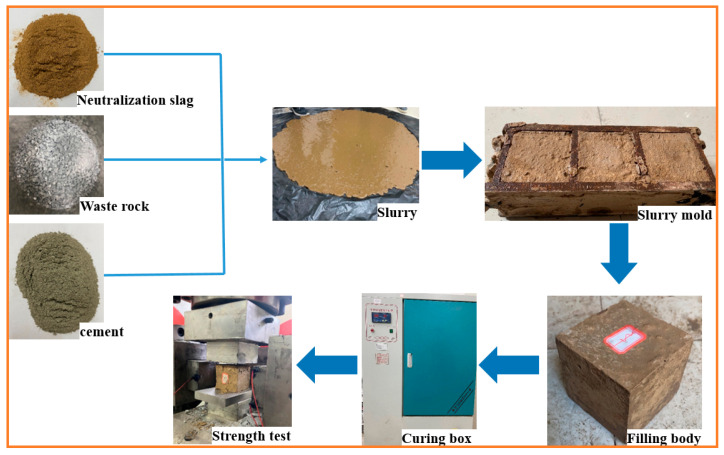
Preparation steps of cemented neutralization slag backfill.

**Figure 5 materials-15-01585-f005:**
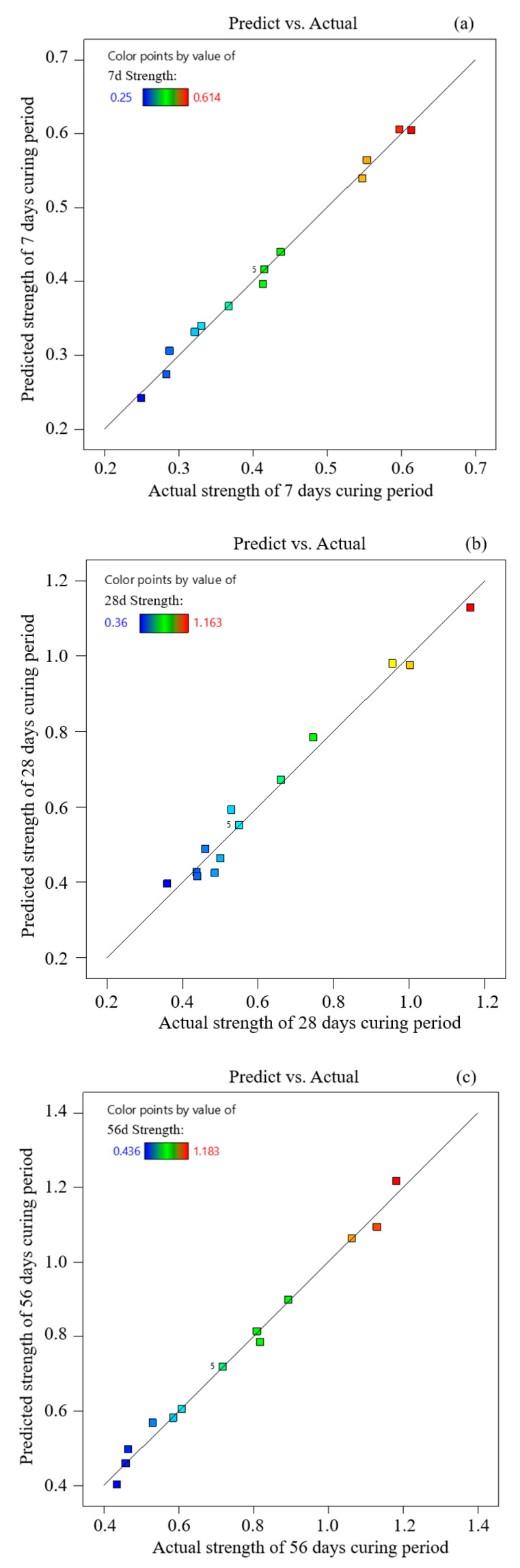
Linear fitting of predicted value and tested value (**a**) curing age at 7 days; (**b**) curing age at 28 days; (**c**) curing age at 56 days.

**Figure 6 materials-15-01585-f006:**
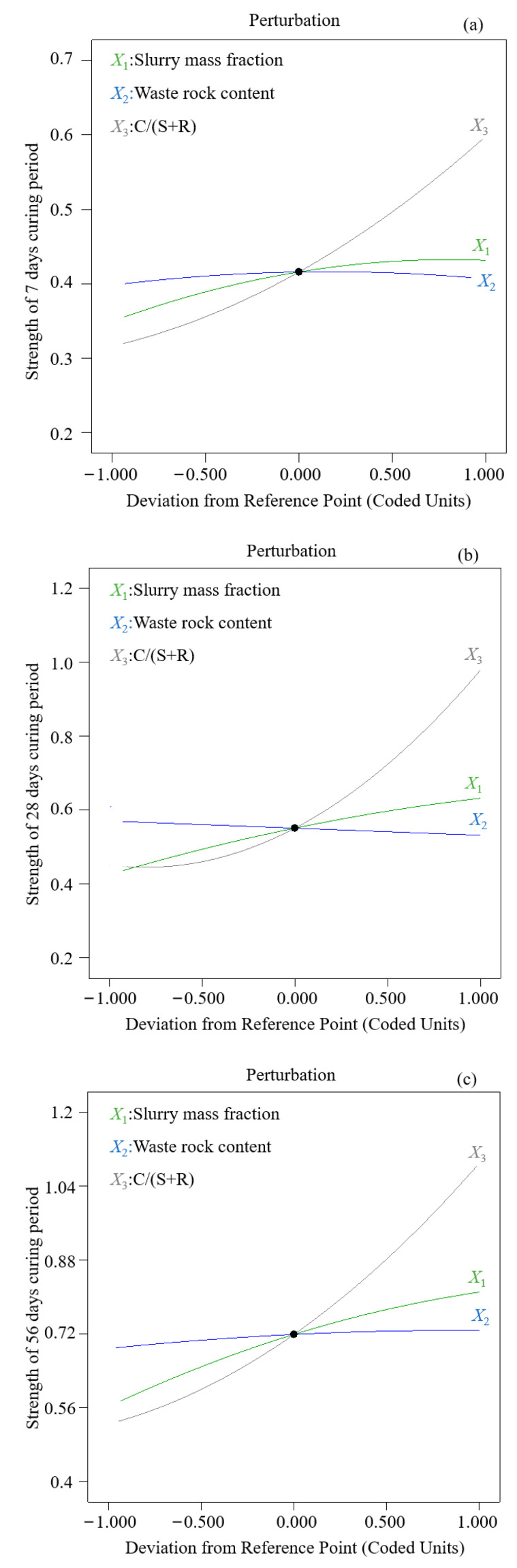
Effect of single factor on uniaxial compressive strength (UCS) of backfill at different curing ages (**a**) curing age at 7 days; (**b**) curing age at 28 days; (**c**) curing age at 56 days.

**Figure 7 materials-15-01585-f007:**
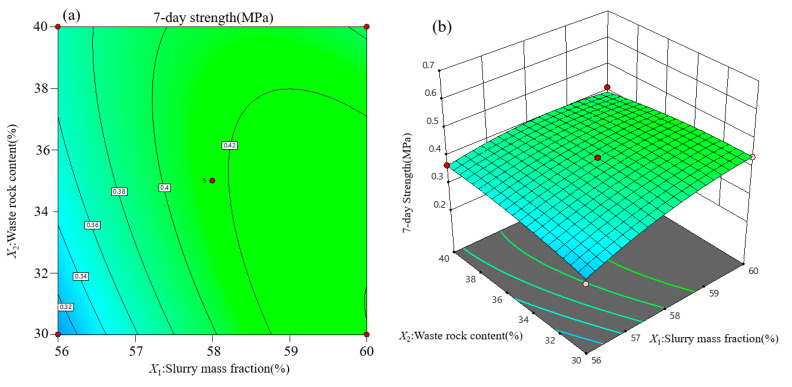
Response surface analysis of influential factors on backfill strength at 7 days: (**a**) contour lines; (**b**) response surface.

**Figure 8 materials-15-01585-f008:**
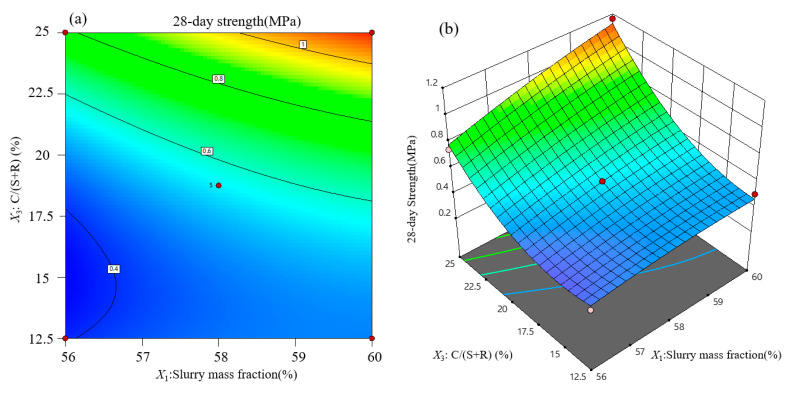
Response surface analysis of influential factors on backfill strength at 28 days: (**a**) contour lines; (**b**) response surface.

**Figure 9 materials-15-01585-f009:**
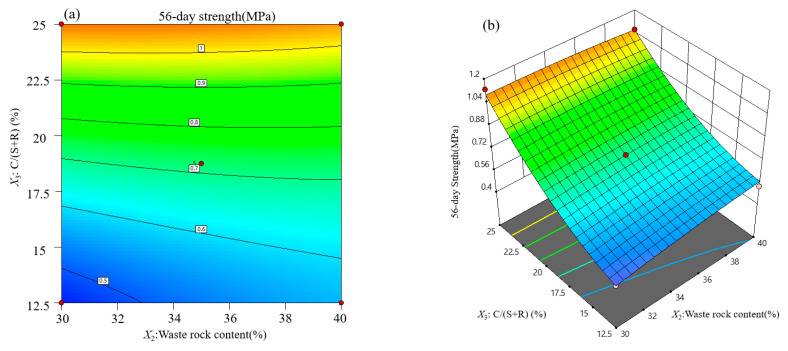
Response surface analysis of influential factors on backfill strength at 56 days: (**a**) contour lines; (**b**) response surface.

**Figure 10 materials-15-01585-f010:**
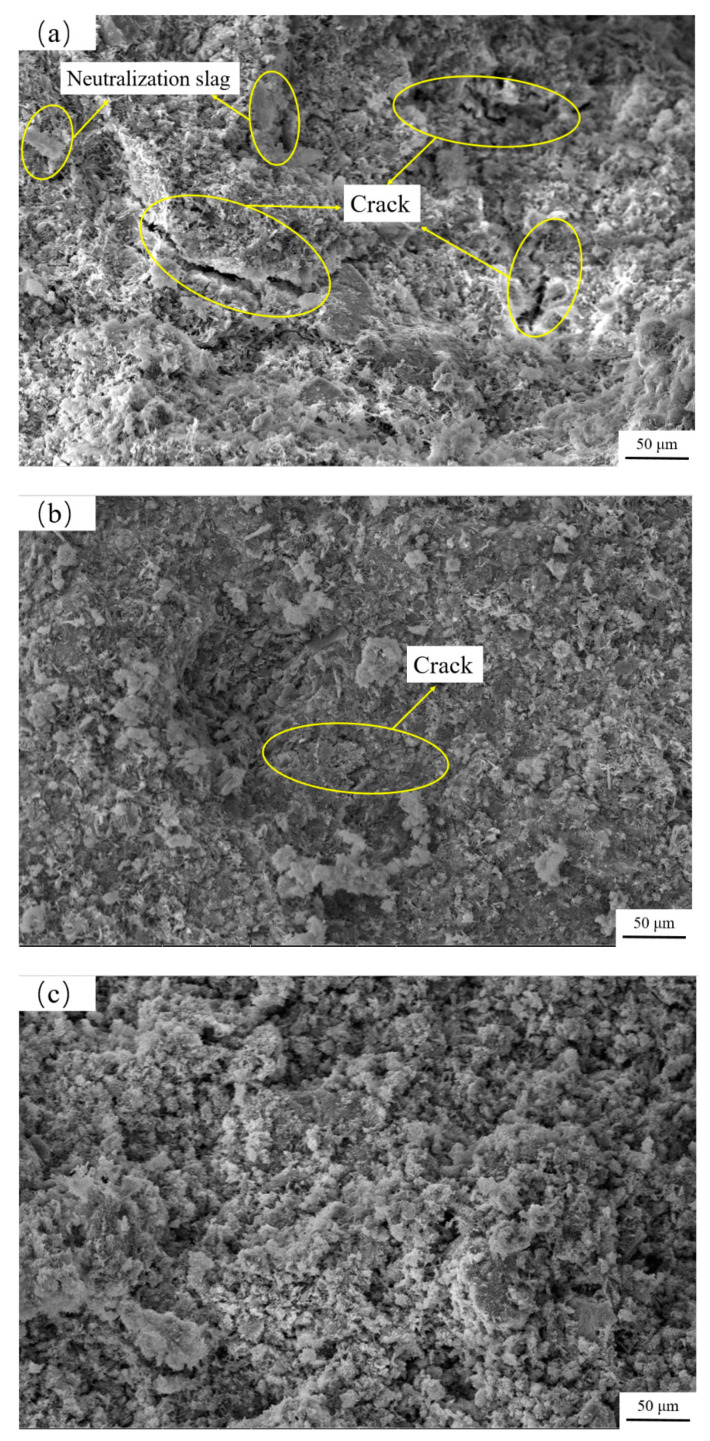
Microstructure of cemented backfill samples: (**a**) curing age at 7 days; (**b**) curing age at 28 days; (**c**) curing age at 56 days.

**Table 1 materials-15-01585-t001:** Physical parameters of neutralization slag.

Aggregate	Density/(g·cm^−3^)	Dense unit weight/(g·cm^−3^)	Porosity/%	Stacking Compactness/%
Neutralization slag	2.418	0.948	0.608	0.392
Waste rock	2.67	1.801	0.33	0.67

**Table 2 materials-15-01585-t002:** Main chemical compositions of neutralization slag.

Component	P_2_O_5_	SO_3_	As	SiO_2_	Ca	Al_2_O_3_	Fe	Others
Content/%	0.26	19.98	0.51	18.05	14.6	4.94	8.23	33.43

**Table 3 materials-15-01585-t003:** Influential factors and level codes of backfill mixture.

Influential Factor	Code Value	Coding Level		
		−1	0	1
Slurry mass fraction (%)	*X* _1_	56	58	60
Waste rock content (%)	*X* _2_	30	35	40
C/(S+R) (%)	*X* _3_	12.5	18.75	25

**Table 4 materials-15-01585-t004:** Box–Behnken design scheme.

Run	Coded Variables			Original Variables		
	*X* _1_	*X* _2_	*X* _3_	Slurry Mass Fraction (%)	Waste Rock Content (%)	C/(S+R) (%)
1	0	0	0	58	35	18.75
2	0	0	0	58	35	18.75
3	0	0	0	58	35	18.75
4	−1	−1	0	56	30	18.75
5	−1	0	−1	56	35	12.5
6	0	0	0	58	35	18.75
7	1	0	1	60	35	25
8	1	0	−1	60	35	12.5
9	1	−1	0	60	30	18.75
10	−1	1	0	56	40	18.75
11	1	1	0	60	40	18.75
12	0	−1	−1	58	30	12.5
13	0	0	0	58	35	18.75
14	−1	0	1	56	35	25
15	0	1	1	58	40	25
16	0	−1	1	58	30	25
17	0	1	−1	58	40	12.5

**Table 5 materials-15-01585-t005:** Test design results and predicted values.

Number	Code Value			Actual Strength/MPa			Predicted Strength/MPa		
	*X* _1_	*X* _2_	*X* _3_	*Y* _1_	*Y* _2_	*Y* _3_	*Y*_1_*	*Y*_2_*	*Y*_3_*
1	0	0	0	0.416	0.551	0.719	0.416	0.551	0.719
2	0	0	0	0.416	0.551	0.719	0.416	0.551	0.719
3	0	0	0	0.416	0.551	0.719	0.416	0.551	0.719
4	−1	−1	0	0.288	0.486	0.465	0.306	0.424	0.498
5	−1	0	−1	0.250	0.360	0.436	0.242	0.395	0.403
6	0	0	0	0.416	0.551	0.719	0.416	0.551	0.719
7	1	0	1	0.598	1.163	1.183	0.606	1.128	1.216
8	1	0	−1	0.331	0.501	0.586	0.340	0.463	0.582
9	1	−1	0	0.438	0.661	0.81	0.440	0.672	0.814
10	−1	1	0	0.368	0.438	0.609	0.366	0.427	0.605
11	1	1	0	0.414	0.53	0.818	0.396	0.592	0.785
12	0	−1	−1	0.284	0.461	0.459	0.274	0.488	0.459
13	0	0	0	0.416	0.551	0.719	0.416	0.551	0.719
14	−1	0	1	0.548	0.746	0.894	0.540	0.784	0.898
15	0	1	1	0.554	1.003	1.064	0.564	0.976	1.064
16	0	−1	1	0.614	0.956	1.131	0.605	0.980	1.094
17	0	1	−1	0.322	0.439	0.531	0.331	0.415	0.568

**Table 6 materials-15-01585-t006:** Accuracy evaluation of response function model.

Source of Variation	Sum of Squares			Mean Square			*F*-Value			*p*-Value		
	*Y* _1_	*Y* _2_	*Y* _3_	*Y* _1_	*Y* _2_	*Y* _3_	*Y* _1_	*Y* _2_	*Y* _3_	*Y* _1_	*Y* _2_	*Y* _3_
Model	0.1877	0.7789	0.8154	0.0209	0.0865	0.0906	113.2	38.34	88.49	<0.0001	<0.0001	<0.0001
*X* _1_	0.0134	0.0851	0.1233	0.0134	0.0851	0.1233	72.54	37.69	120.39	<0.0001	0.0005	<0.0001
*X* _2_	0.0001	0.003	0.0031	0.0001	0.003	0.0031	0.7843	1.31	3.01	0.4052	0.2895	0.1264
*X* _3_	0.1588	0.5549	0.6384	0.1588	0.5549	0.6384	861.69	245.82	623.6	<0.0001	<0.0001	<0.0001
*X* _1_ *X* _2_	0.0027	0.0017	0.0046	0.0027	0.0017	0.0046	14.68	0.7629	4.52	0.0065	0.4114	0.0712
*X* _1_ *X* _3_	0.0002	0.019	0.0048	0.0002	0.019	0.0048	1.3	8.44	4.72	0.291	0.0228	0.0664
*X* _2_ *X* _3_	0.0024	0.0012	0.0048	0.0024	0.0012	0.0048	13.03	0.5273	4.72	0.0086	0.4913	0.0664
*X* _12_	0.0027	0.0021	0.0044	0.0027	0.0021	0.0044	14.71	0.9234	4.34	0.0064	0.3686	0.0756
*X* _22_	0.0008	0	0.0005	0.0008	0	0.0005	4.24	0	0.4976	0.0784	1	0.5033
*X* _32_	0.0071	0.1129	0.0328	0.0071	0.1129	0.0328	38.65	50.01	32.03	0.0004	0.0002	0.0008
Residual	0.0013	0.0158	0.0072	0.0002	0.0023	0.001						
Lack of Fit	0.0013	0.0158	0.0072	0.0004	0.0053	0.0024						
Pure Error	0	0	0	0	0	0						
Total	0.189	0.7947	0.8226									

**Table 7 materials-15-01585-t007:** Calculation model solving results with C++ program language.

Solve Count	M_a_	M_s_	M_j_	M_w_	*X* _1_	*X* _2_	*X* _3_	*Y* _1_	*Y* _2_	*Y* _3_
0	0.46	0.27	0.18	0.64	58.7	36.8	24.9	0.59	1.01	1.12
1	0.46	0.27	0.18	0.65	58.3	36.8	24.9	0.59	0.99	1.09
2	0.46	0.29	0.15	0.64	58.6	39.0	20.0	0.44	0.62	0.81
3	0.46	0.29	0.18	0.63	59.7	39.0	24.0	0.53	0.99	1.09
4	0.46	0.29	0.18	0.64	59.4	39.0	24.0	0.54	0.97	1.07
5	0.48	0.24	0.18	0.64	58.6	33.2	24.9	0.61	1.02	1.14
6	0.48	0.24	0.18	0.65	58.2	33.2	24.9	0.6	0.99	1.11
7	0.48	0.27	0.15	0.64	58.5	35.6	20.1	0.45	0.64	0.81
8	0.48	0.27	0.18	0.63	59.7	35.6	24.1	0.57	1.01	1.12
9	0.48	0.27	0.18	0.64	59.3	35.6	24.1	0.57	0.98	1.1
10	0.48	0.29	0.15	0.63	59.6	37.8	19.4	0.43	0.63	0.82
11	0.48	0.29	0.15	0.64	59.2	37.8	19.4	0.43	0.62	0.81
12	0.51	0.24	0.15	0.64	58.4	32.2	20.1	0.46	0.65	0.81
13	0.51	0.24	0.18	0.63	59.6	32.2	24.1	0.6	1.03	1.15
14	0.51	0.24	0.18	0.64	59.2	32.2	24.1	0.59	1	1.13
15	0.51	0.27	0.15	0.63	59.5	34.5	19.4	0.45	0.66	0.83
16	0.51	0.27	0.15	0.64	59.2	34.5	19.4	0.45	0.64	0.81
17	0.53	0.24	0.15	0.63	59.5	31.2	19.5	0.45	0.68	0.83
18	0.53	0.24	0.15	0.64	59.1	31.2	19.5	0.45	0.66	0.81

**Table 8 materials-15-01585-t008:** Error analysis of confirmatory test.

	Curing Time	7-Day	28-Day	56-Day
Backfill Strength				
Estimate value		0.46	0.65	0.81
Test value		0.42	0.64	0.85
Error value		8.7%	1.54%	4.94%

## Data Availability

The data presented in this study are available on request from the corresponding author.

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
