# Peer review of "Multi-Objective Function Optimization of Cemented Neutralization Slag Backfill Strength Based on RSM-BBD"

_materials, 2022, doi:10.3390/ma15041585_

Round 1
Reviewer 1 Report
The authors have done work on Multi objective function optimization of cemented neutralization slag backfill strength based on RSM-BBD. Here are few suggestions that can be incorporated in the revised manuscript before final publication:
- How the particle size of the used materials was measured? Which technique/equipment was used?
- Write the complete process in detail about the particle size distribution range of slag.
- How the density of slag was measured? Which technique/equipment was used?
- Why there is a shuffling in a curve of Particle size composition of waste rock filled with aggregate in Fig.2 between 0 to 1.
- What is the purpose of adding Measurement of slurry expansion with mass fraction of 64%.figure.3?
- Which relationships were involved in the fitting quality of the regression model ?
- Equations 2, 3 and 4 should be included in the table separately.
Reviewer 2 Report
Dear Authors,
The work is interesting, it brings new scientific and practical knowledge. It will surely arouse the interest of readers.
However, please clarify the following points and complete them in the text.
Chapter 2.1. Materials
Fig. 1. and 2.
These are dependencies of Y (X), so they can be described by their function. Such presentation of the results, in this case the dependence of fractional content (Y) on particle size (X), is valid only for specific conditions, e.g. type of crusher, grinding method and parameters etc. Therefore, I believe that it is correct to show the results in the form of bar graphs. If the form of the graphs were to be as in the manuscript, the points should not be combined and the data that determined such a specific fractional distribution should be very precisely described.
Chapter 2.4. Statistical analysis
- There is no justification as to why the CMS statistical model was used. What is the applicability of this model?
- Equation 1
There is no explanation for the coefficients a0, ai, aii, aij. How are these coefficients selected for the analyzed case (for a three-component mixture)?
Chapter 3.1. Evaluation of strength test and model fitting
- Please explain why functions 2, 3, 4 have a defined sequence? What are the values of the coefficients, e.g. in the equation 1: 29.32637, 0.858937, 0.204500 ... etc. - similar in equations 2, 3, 4?
- Table 6
What is the difference of F value to P value?
Chapter 3.2. Influence of single factor on backfill strength
Fig. 6.
What are the reasons for the differences in the function sequence? What are the different points of intersection?
Chapter 3.4. SEM analysis of backfill at different curing ages
Fig. 10.
The causes of crack formation and their disappearance require a broader, explanatory comment.
Chapter 3.5. Backfill strength optimization and verification
Please explain why the model described by equations 5 - 9 was used for optimization? What are the features of this model that indicate its use?
Best regards,
Reviewer
Round 2
Reviewer 1 Report
The paper can be accepted in current form